# Comparison of delivery outcomes in low-dose and high-dose oxytocin regimens for induction of labor following cervical ripening with a balloon catheter: A retrospective observational cohort study

Heidi Kruit[1]☯*, Irmeli Nupponen[2‡], Seppo Heinonen[1‡], Leena Rahkonen[1]☯

1 Department of Obstetrics and Gynecology, University of Helsinki and Helsinki University Hospital, Helsinki, Finland, 2 Department of Neonatology, University of Helsinki and Helsinki University Hospital, Helsinki, Finland

☯ These authors contributed equally to this work.
‡ IN and SH also contributed equally to this work.
* heidi.kruit@hus.fi

**Data Availability Statement:** Data cannot be shared publicly because of institutional policy of

## Abstract

A variety of oxytocin regimens are used for labor induction and augmentation. Considering the increasing rates of labor induction, it is important to assess the most optimal oxytocin regimen without compromising maternal and fetal safety. The aim of this study was to compare delivery outcomes of low-dose and high-dose oxytocin induction protocols. This retrospective cohort study of 487 women comparing low-dose oxytocin protocol (n = 280) and high-dose oxytocin protocol (n = 207) in labor induction following cervical ripening by balloon catheter was performed in Helsinki University Hospital after implementation of a new oxytocin induction protocol. The study included two six-month cohorts from 2016 and 2019. Women with vital singleton pregnancies ≥37 gestational weeks, cephalic presentation, and intact amniotic membranes were included. The primary outcome was the rate of vaginal delivery. The secondary outcomes were the rates of maternal and neonatal infections, post-partum hemorrhage, umbilical artery blood pH-value, admission to neonatal intensive care, and induction-to-delivery interval. Statistical analyses were performed by using IBM SPSS Statistics for Windows (Armonk, NY, USA). The rate of vaginal delivery was higher [69.9% (n = 144) vs. 47.9% (n = 134); p<0.004] and the rates of maternal and neonatal infection were lower during the new high-dose oxytocin protocol [maternal infections 13.6% (n = 28) vs. 22.1% (n = 62); p = 0.02 and neonatal infection 2.9% (n = 6) vs. 14.6% (n = 41); p<0.001, respectively]. The rates of post-partum hemorrhage, umbilical artery blood pH-value <7.05 or neonatal intensive care admissions did not differ between the cohorts. The median induction-to-delivery interval was shorter in the new protocol [32.0 h (IQR 18.5–42.7) vs. 37.9 h (IQR 27.8–52.8); p<0.001]. In conclusion, implementation of the new continuous high-dose oxytocin protocol resulted in higher rate of vaginal delivery and lower rate of maternal and neonatal infections. Our experience supports the use of high-dose continuous oxytocin induction regimen with a practice of stopping oxytocin once active labor is achieved, and a

the Institutional Review board of Helsinki University Hospital (Helsinki and Uusimaa Hospital District Committee for Obstetrics and Gynecology). Data are available from the Institutional Review board of Helsinki University Hospital (contact via tutkimusneuvonta@hus.fi or via corresponding author heidi.kruit@hus.fi) for researchers who meet the criteria for access to confidential data.

**Funding:** The author(s) received no specific funding for this work.

**Competing interests:** The authors have declared that no competing interests exist.

15–18-hour maximum duration for oxytocin induction in the latent phase of labor following cervical ripening with a balloon catheter.

## Introduction

Oxytocin, a hormone secreted from the posterior pituitary, causes uterine contractions [1]. Oxytocin levels gradually increase during pregnancy and labor, and a four-fold rise of oxytocin levels occurs during vaginal delivery [2]. Oxytocin is the most common agent used for induction and augmentation of labor, administered by intravenous infusion according to uterine contraction frequency.

Currently, there is a wide variation in oxytocin regimens by administration with either continuous or pulsed infusions, linear or non-linear incremental increases, and high-dose or low-dose regimens [3]. The optimal timing, dose and infusion type are still being debated over. High-dose oxytocin regimens may cause hyperstimulation associated with adverse maternal and neonatal outcomes, while low-dose protocols may prolong the induction to delivery interval increasing the chance of maternal and neonatal infections [4, 5]. Evaluation of the best oxytocin regimen in induction of labor is challenging, as labor induction is influenced by various antenatal factors, such as parity, maternal height and weight, prior mode of delivery, cervical ripeness classified by Bishop's score, fetal weight, and gestational age [6, 7]. Several studies on evidence-based labor management and induction of labor suggest that combination methods such as Foley catheter and either misoprostol or oxytocin are associated with a shorter labor durations than one method alone [8–10]. It is important to determine whether high-dose oxytocin regimens can improve successful induction of labor without compromising maternal and fetal safety. Oxytocin has usually been assessed as an isolated intervention, and as different combination methods for labor induction become more popular, further research on oxytocin use alone and in combination with other methods in different regimens is warranted.

The aim of this study was to compare the delivery outcomes of two different oxytocin protocols following cervical ripening by balloon catheter in the same academic tertiary hospital following implementation of a new high-dose oxytocin protocol for labor induction.

## Material and methods

This retrospective cohort study comparing low-dose and high-dose oxytocin protocols in induction of labor was performed in the Department of Obstetrics and Gynecology in Helsinki University Hospital, with approximately 13 500 deliveries annually. The rate of labor induction is 30%, and the overall rate of cesarean delivery is 20%. In Helsinki University Hospital, balloon catheter is the principal method used for cervical ripening, and in approximately 50% of the women labor induction is continued by amniotomy and oxytocin infusion.

The study included two six-month cohorts of women with vital singleton pregnancies at or beyond 37 gestational weeks, cephalic presentation, and intact amniotic membranes, undergoing induction of labor with cervical ripening by balloon catheter followed by amniotomy and oxytocin. The first cohort was obtained during the old oxytocin induction protocol ("old protocol") between January 1st and June 30th 2016, and the second cohort was obtained during the new oxytocin induction protocol ("new protocol") between January 1st and June 30th 2019. Women with multiple gestation, breech presentation, stillbirth, pre-labor rupture of membranes, preterm pregnancy, and ripe cervix with Bishop score > 6 were excluded from the study. Also, women who did not need oxytocin infusion during their labor induction were

excluded. The potential bias of the study were the retrospective design and the decision to include only women undergoing induction of labor at term, rather than at any time during the third trimester. This bias was addressed by choosing two identical cohorts within two identical time periods during the two different oxytocin induction protocols. The study protocol was approved by the institutional review board (IRB) of the hospital region (Helsinki and Uusimaa Hospital District Committee for Obstetrics and Gynecology, HUS/3172/2018 and HUS/54/2019). Written informed consent was waived by the IRB according to national legislation (Medical Research Act 488/1999, chapter 2 a (23.4.2004/295), section 5 and 10a) and the retrospective nature of the study. All data were fully anonymized prior to access and the analyses. The study was performed in accordance with the Declaration of Helsinki and reported in compliance with the STROBE statement.

The primary outcome was the rate of vaginal delivery. The secondary outcomes were the rates of maternal and neonatal infections, postpartum hemorrhage, umbilical artery blood gas pH-value, admission to neonatal intensive care, and induction to delivery interval. The data on baseline characteristics and delivery outcomes were obtained from the hospital database.

Induction of labor was started by cervical ripening with a single 60–80 ml balloon catheter (Rüsch 2-way Foley Couvelaire tip catheter size 22 Ch, Teleflex Medical, Athlone, Ireland). Light traction was applied by taping the catheter on the inner thigh. The balloon was retained for a maximum of 24 hours. When Bishop score $\geq$ 6 was reached, labor induction was continued by amniotomy and oxytocin induction if spontaneous contractions and onset of labor had not occurred.

Helsinki University Hospital used the old low-dose intermittent oxytocin protocol in the induction of labor until January 2017, after which it was replaced by the new high-dose continuous oxytocin protocol. The oxytocin protocols are described in Tables 1 and 2.

In the old oxytocin protocol, oxytocin infusion was started 2 hours after amniotomy in the absence of spontaneous regular contractions. Oxytocin (Syntocinon® 8,3 μg [5 IU]) diluted in 500 ml of NaCl 0.9%) infusion was started at 2.5 mIU/min (15 ml/h), and the dose was

**Table 1. The old intravenous low-dose oxytocin infusion protocol.**

| Concentration: Oxytocin (Syntocinon®) 5 IU diluted in 500 ml of 0.9% saline (0,01 IU/ml) | | | | |
|---|---|---|---|---|
| Time; hours:minutes | Dose mIU/min | Dose ml/h | Total Oxytocin units infused (mIU) | Total Volume infused (ml) |
| 0:30 | 2,5 | 15 | 75 | 7.5 |
| 1:00 | 4,2 | 25 | 200 | 20 |
| 1:30 | 5,8 | 35 | 375 | 37.5 |
| 2:00 | 7,5 | 45 | 600 | 60 |
| 2:30 | 9,2 | 55 | 875 | 87.5 |
| 3:00 | 10,8 | 65 | 1200 | 120 |
| 3:30 | 12,5 | 75 | 1575 | 157.5 |
| 4:00 | 14,2 | 85 | 2000 | 200 |
| 4:30 | 15,0 | 90 | 2450 | 245 |
| 5:00 | 15,0 | 90 | 2900 | 290 |
| 5:30 | 15,0 | 90 | 3350 | 335 |
| 6:00 | 15,0 | 90 | 3800 | 380 |

Break 2–6 hours, after which another 6-hour infusion repeated as above

Maximum dose: 15 mIU/min

Maximum continuous duration: 6 hours

Total maximum dose infused: 3.8 IU x 2 = 7.6 IU

**Table 2. The new intravenous high-dose oxytocin infusion protocol.**

| Oxytocin (Syntocinon®) 8,3 μg = 5 IU diluted in 500 ml of 0.9% saline (0,01 IU/ml) | | | | |
|---|---|---|---|---|
| Time; hours:minutes | Dose mIU/min | Dose ml/h | Total Oxytocin units infused (mIU) | Total Volume infused (ml) |
| 00:20 | 2,5 | 15 | 50 | 5.0 |
| 00:40 | 4,2 | 25 | 133 | 13.3 |
| 01:00 | 5,8 | 35 | 250 | 25.0 |
| 01:20 | 7,5 | 45 | 400 | 40.0 |
| 01:40 | 9,2 | 55 | 583 | 58.3 |
| 02:00 | 10,8 | 65 | 800 | 80.0 |
| 02:20 | 12,5 | 75 | 1050 | 105.0 |
| 02:40 | 14,2 | 85 | 1333 | 133.3 |
| 03:00 | 15,8 | 95 | 1650 | 165.0 |
| 03:20 | 17,5 | 105 | 2000 | 200.0 |
| 03:40 | 19,2 | 115 | 2383 | 238.3 |
| 04:00 | 20,0 | 120 | 2783 | 278.3 |
| 04:20 | 20,0 | 120 | 3183 | 318.3 |
| 04:40 | 20 | 120 | 3583 | 358.3 |
| 05:00 | 20 | 120 | 3983 | 398.3 |
| 05:20 | 20 | 120 | 4383 | 438.3 |
| 05:40 | 20 | 120 | 4783 | 478.3 |
| 06:00 | 20 | 120 | 5183 | 518.3 |
| 06:20 | 20 | 120 | 5583 | 558.3 |
| 06:40 | 20 | 120 | 5983 | 598.3 |
| 07:00 | 20 | 120 | 6383 | 638.3 |
| 07:20 | 20 | 120 | 6783 | 678.3 |
| 07:40 | 20 | 120 | 7183 | 718.3 |
| 08:00 | 20 | 120 | 7583 | 758.3 |
| 08:20 | 20 | 120 | 7983 | 798.3 |
| 08:40 | 20 | 120 | 8383 | 838.3 |
| 09:00 | 20 | 120 | 8783 | 878.3 |
| 09:20 | 20 | 120 | 9183 | 918.3 |
| 09:40 | 20 | 120 | 9583 | 958.3 |
| 10:00 | 20 | 120 | 9983 | 998.3 |
| 10:20 | 20 | 120 | 10383 | 1038.3 |
| 10:40 | 20 | 120 | 10783 | 1078.3 |
| 11:00 | 20 | 120 | 11183 | 1118.3 |
| 11:20 | 20 | 120 | 11583 | 1158.3 |
| 11:40 | 20 | 120 | 11983 | 1198.3 |
| 12:00 | 20 | 120 | 12383 | 1238.3 |

Maximum dose: 20 mIU/min

Maximum continuous duration: 12(-18) hours

Total dose infused: 12.38 UI

increased by 1.7 mIU/min (10 ml/h) increments in every 30 minutes up to 15 mIU/min (90 ml/h) (Table 1). Oxytocin infusion was continuously administered for 6 hours, with the total maximum dose of 3.8 IU was infused, followed by a break of 2–6 hours and another 6 hours of oxytocin infusion, or until contractions deemed adequate or active labor was reached.

Oxytocin was altogether administered for 12–18 hours, with a break of 2–6 hours following each six-hour infusion, after which induction was considered failed.

In the new continuous high-dose oxytocin protocol, Oxytocin (Syntocinon® 8,3 μg (5 IU)) diluted in 500 ml of NaCl 0.9%) infusion was started 1–2 hours following amniotomy with 2.5 mIU/min (15 ml/h) and the dose was increased by 1.7 mIU/min (10 ml/h) increments in every 20 minutes up to 20 mIU/min (120ml/h) or until contractions were deemed adequate (Table 2). When contractions were deemed adequate and active labor was reached, oxytocin was paused for 1–2 hours, and administration was later restarted for augmentation of labor if necessary. Oxytocin induction was continued continuously for a maximum of 12–18 hours, with the total maximum dose of 12.38 IU, after which induction of labor was deemed failed and cesarean section was performed [11].

Contractions were deemed adequate when 3–5 contractions occurred in 10-minute period. Active labor was defined as adequate contractions and cervical dilation of 6 cm or more. Continuous cardiotocography during oxytocin induction and labor was routinely used. GBS was routinely tested in all women by a rapid qualitative in vitro GBS test (Xpert® GBS, Cepheid, Sunnyvale, California, USA) at the time of admission. Administration of prophylactic Benzyl-penicillin with the first dose of 4 million units intravenously, followed by 2.5 million units every 4 hours until delivery was routinely used for prophylaxis in GBS-positive women. In case of a penicillin allergy, clindamycin 900 mg was administered every 8 hours intravenously.

Maternal infections were categorized as intrapartum (during labor) and postpartum (within one week from delivery). The criteria for intrapartum infection were maternal fever $\geq$ 38˚C during labor and at least one of the following criteria: fetal tachycardia $\geq$ 160 bpm, uterine tenderness, purulent amniotic fluid or vaginal discharge, total white cell count > 15E9/L. Postpartum infections included endometritis, cesarean or episiotomy wound infection, deep abdominal or pelvic infection, sepsis, and puerperal fever of unknown origin. Neonatal infections were categorized by a neonatologist into blood culture positive sepsis, clinical sepsis, and suspected sepsis. Neonatal clinical sepsis was defined as blood culture negative infection with symptoms and signs consistent with sepsis (respiratory distress, apnea, tachycardia, poor perfusion, low blood pressure, fever, hypoglycemia or hyperglycemia, irritability, feeding problems, lethargy, convulsions), abnormal blood values (C-reactive protein (>20 mg/l), leukocytosis or leucopenia, increased neutrophil precursors and thrombocytopenia), and positive reaction to a minimum of five-day antibiotic treatment. The cases of suspected sepsis had at least one symptom and at least one abnormal laboratory test value, and a positive response to antibiotic treatment.

Statistical analyses were performed by using IBM SPSS Statistics for Windows, Version 26.0 (Armonk, NY, USA). Categorical variables were analyzed for odds ratios (OR) with 95% confidence interval (CI). Categorical variables were compared by the chi-square test and Fisher's exact test when appropriate. Data with continuous variables were performed by T-test when the data followed normal distribution and by a Mann-Whitney U test if the data did not follow normal distribution. Univariate and multivariate logistic regression analyses were performed to assess relative risk for intrapartum infection. Variables used in the multivariate analyses were parity, maternal age, height, body mass index (BMI), previous cesarean delivery, Bishop score, gestational age, indication for labor induction, duration of oxytocin infusion, and the induction to delivery interval $\geq$ 48 h. A p-value <0.05 was considered statistically significant.

## Results

We identified 3290 and 3085 deliveries during the selected time periods of the old and new oxytocin induction protocol, examined 743 and 853 women who underwent induction of

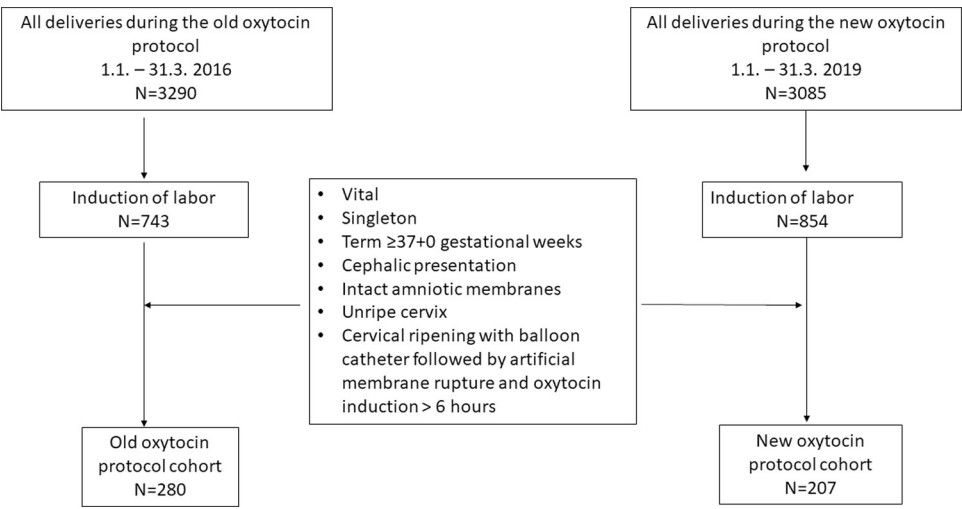

**Fig 1. Flowchart of the study population.**

labor for eligibility, and confirmed 280 and 207 women eligible, respectively (Fig 1). A total of 487 women with the mean age of 32.1 (5.2 SD) years, the mean BMI 26.7 (6.0 SD) and the median gestational age of 41.3 (range 39.6–42.0) weeks were included in the study. The old protocol included 280 women (57.5%) and the new protocol included 207 (42.5%) women (Fig 1). The characteristics of the study population are presented in Table 3. Women in the new protocol were older, more often had pregestational diabetes, and lower Bishop score at the start of labor induction, while there were more post-term pregnancies in the old protocol

**Table 3. Characteristics of the study population.**

| | Old protocol | | New protocol | | p-value |
|---|---|---|---|---|---|
| | **n = 280** | **(%)** | **n = 207** | **(%)** | |
| Primiparous | 210 | 75.0 | 140 | 68.0 | 0.09 |
| Maternal age, mean (SD) | 31.1 | 5.3 | 33.3 | 4.9 | <0.001 |
| Age ≥37 years | 45 | 16.1 | 53 | 25.7 | 0.01 |
| BMI, mean (SD) | 26.4 | 5.5 | 25 | 6.7 | 0.26 |
| BMI ≥35 | 25 | 8.9 | 23 | 11.2 | 0.42 |
| IVF | 23 | 8.2 | 24 | 11.7 | 0.21 |
| Smoking | 14 | 5.0 | 10 | 4.9 | 0.94 |
| Post-term ≥ 41 weeks | 48 | 17.7 | 13 | 6.3 | <0.001 |
| Pregestational DM | 2 | 0.7 | 11 | 5.3 | 0.002 |
| Gestational diabetes | 94 | 33.6 | 70 | 34.0 | 0.93 |
| Medicated gestational diabetes | 24 | 8.6 | 24 | 11.7 | 0.26 |
| Maternal height < 164 cm | 117 | 41.8 | 68 | 33.2 | 0.05 |
| Bishop < 3 | 114 | 40.7 | 104 | 50.5 | 0.03 |
| GBS-colonization | 63 | 23.2 | 44 | 21.4 | 0.64 |
| **Indication for labor induction** | | | | | |
| Post-term pregnancy | 136 | 48.6 | 87 | 42.2 | 0.17 |
| Diabetes | 48 | 17.1 | 46 | 22.3 | 0.15 |
| Hypertension or preeclampsia | 28 | 10.0 | 20 | 9.7 | 0.92 |
| Other | 68 | 24.4 | 53 | 25.7 | 0.72 |

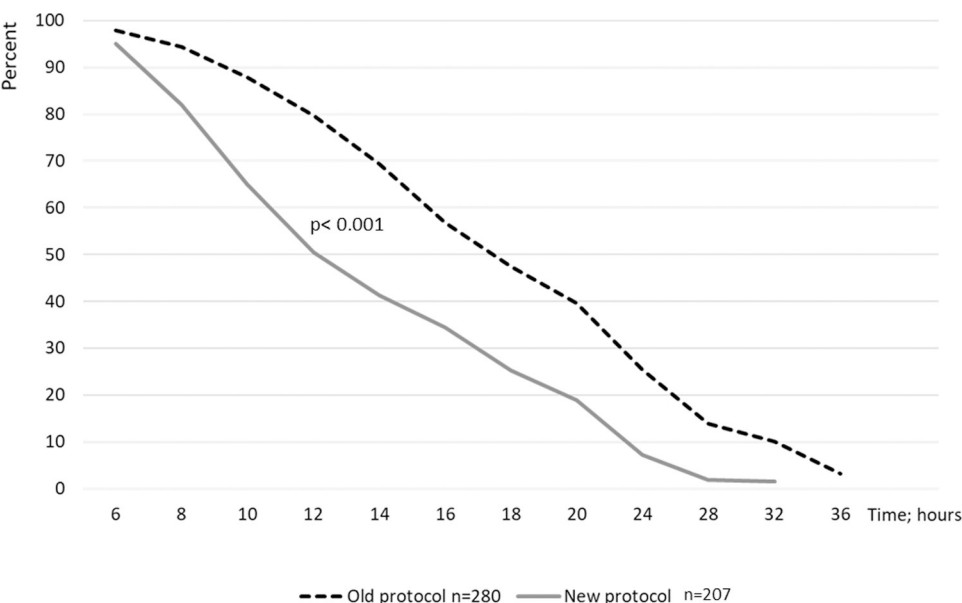

**Fig 2. Duration of oxytocin induction in the old and new protocols.**

cohort (Table 3). The indications for labor induction did not differ between the groups (Table 3).

The median duration of oxytocin induction, including the possible breaks during the infusion, was 12.1 (range 8.6–18.1) hours in the new protocol, and 17.4 (range 13.1–24.4) hours in the old protocol; p< 0.001 (Fig 2). Active labor was more often reached in the new protocol compared to the old protocol [80.1% (n = 165) vs. 64.3% (n = 180); p <0.001]. The median induction to delivery interval was significantly shorter in the new protocol [32.0 (IQR 18.5–42.7) hours vs. 37.9 (IQR 27.8–52.8) hours; p<0.001]. The duration of active labor did not differ between the protocols [9.0 (IQR 6.3–12.9) hours vs. 9.5 (IQR 6.6–13.4) hours; p = 0.50].

The maternal and neonatal delivery outcomes are presented in Table 4. The rate of vaginal delivery was higher in the new protocol [69.9% (n = 144) vs. 47.9% (n = 134); p<0.004]. There were more cesarean sections performed due to fetal distress and failed labor induction during the old protocol (Table 4). The rates of maternal and neonatal infection were lower during the new protocol [maternal infections 13.6% (n = 28) vs. 22.1% (n = 62); p = 0.02 and neonatal infection 2.9% (n = 6) vs. 14.6% (n = 41); p< 0.0001, respectively, Table 4]. In the old protocol, maternal or neonatal infection occurred in 29.3% (n = 82) of the labors, while in the new protocol the corresponding rate was 14.6% (n = 30); p<0.001. The rates of vaginal delivery, cesarean delivery, maternal and neonatal infections, and the respective durations of oxytocin induction are presented in Fig 3. The mean birthweight, the rates of post-partum hemorrhage and umbilical artery blood pH-value <7.05 did not differ between the cohorts (Table 4).

Nulliparity OR 2.4 (95%CI 1.3–4.4), Bishop score < 3 OR 1.7 (95%CI 1.1–2.7), oxytocin infusion of more than 12 hours OR 2.9 (95%CI 1.6–5.2) and prolonged induction to delivery interval of 48 hours or more OR 2.9 (95%CI 1.8–4.7) were associated with maternal infection. After adjustment with parity, age, height, BMI, and induction indication, oxytocin infusion of >12 hours OR 2.4 (95%CI 1.2–4.7) and prolonged induction to delivery interval of 48 hours OR 2.3 (95% CI 1.3–4.1) remained significant. Neonatal infection was, after adjustment for parity, maternal age, BMI, height, and gestational age, associated with oxytocin infusion > 12

**Table 4. Delivery outcomes.**

| | Old protocol | | New protocol | | p-value |
|---|---|---|---|---|---|
| | n = 280 | (%) | n = 207 | (%) | |
| Vaginal delivery | 134 | 47.9 | 144 | 69.9 | <0.004 |
| Instrumental vaginal birth | 35 | 12.5 | 32 | 15.5 | 0.44 |
| Cesarean section | | | | | |
| Fetal distress | 28 | 10 | 9 | 4.4 | 0.02 |
| Failed induction (cx <6cm) | 71 | 25.4 | 33 | 16.0 | 0.01 |
| Labor dystocia (cx >6cm) | 27 | 9.6 | 18 | 8.7 | 0.73 |
| Other | 20 | 7.1 | 2 | 1.0 | 0.001 |
| III -IV grade perineal tear | 4 | 1.4 | 4 | 1.9 | 0.66 |
| Placental retention | 14 | 5 | 4 | 1.9 | 0.08 |
| Post-partum haemorrhage > 1000ml | 61 | 21.8 | 59 | 28.6 | 0.08 |
| Induction to delivery interval <24 h | 51 | 18.2 | 74 | 35.9 | <0.001 |
| Induction to delivery interval <48 h | 189 | 67.5 | 168 | 81.6 | 0.001 |
| Fetal scalp blood sampling | 77 | 27.5 | 13 | 6.3 | <0.001 |
| Birthweight [mean (SD)] | 3673 | 487 | 3674 | 470 | 0.88 |
| Birthweight ≥+2 SD | 6 | 2.1 | 10 | 4.9 | 0.10 |
| Apgar 5min <7 | 13 | 4.8 | 9 | 4.4 | 0.83 |
| Umbilical artery pH <7.05 | 2 | 0.7 | 0 | | 0.51 |
| Umbilical artery BE <-12.0 | 5 | 1.9 | 3 | 1.5 | 1 |
| Maternal infection | 62[1] | 22.1 | 28 | 13.6 | 0.02 |
| Intrapartum infection | 46 | 16.4 | 22 | 10.7 | 0.07 |
| Postpartum infection | 18 | 6.4 | 6 | 2.9 | 0.08 |
| Neonatal infection | 41 | 14.6 | 6 | 2.9 | <0.001 |
| Maternal or neonatal infection | 82 | 29.3 | 30 | 14.6 | <0.001 |
| Maternal and neonatal infection | 21 | 7.5 | 5 | 2.4 | 0.01 |
| Neonatal intensive care unit admission | 24 | 8.6 | 23 | 11.2 | 0.34 |

[1]Two women who had an intrapartum infection presented with a separate post-partum cesarean wound infection later after being discharged from the hospital. Both women had antibiotic treatment and revision surgery.

hours OR 2.7 (95% CI 1.1–6.9) and induction to delivery interval > 48 hours OR 2.2 (95% CI 1.1–4.6).

After implementation of the new continuous oxytocin protocol, the number of emergency cesarean deliveries following induction of labor have decreased, while the rate of labor induction is increasing (Fig 4) [12].

## Discussion

In Helsinki University hospital, two different oxytocin regimens for labor induction were used over the past five years; the old protocol with high starting and low continuing dose of 6-hour intermittent infusion, and a new protocol with high starting dose and continuous 12–18 –hour infusion. During the old protocol, breaks of 2–6 hours between each six-hour intermittent infusion were applied, prolonging the induction to delivery interval. In addition to higher dose, the new protocol included a change to continuous infusion, instituting a practice of stopping oxytocin once active labor is achieved, and instituting a maximum duration of induction before it was considered to have failed. The rate of vaginal delivery was higher (70% vs. 48%; p<0.004), and the rates of maternal and neonatal infections were lower (14% vs. 22%; p = 0.02 and 3% vs. 15%; p<0.001, respectively) during the new protocol. After the implementation of

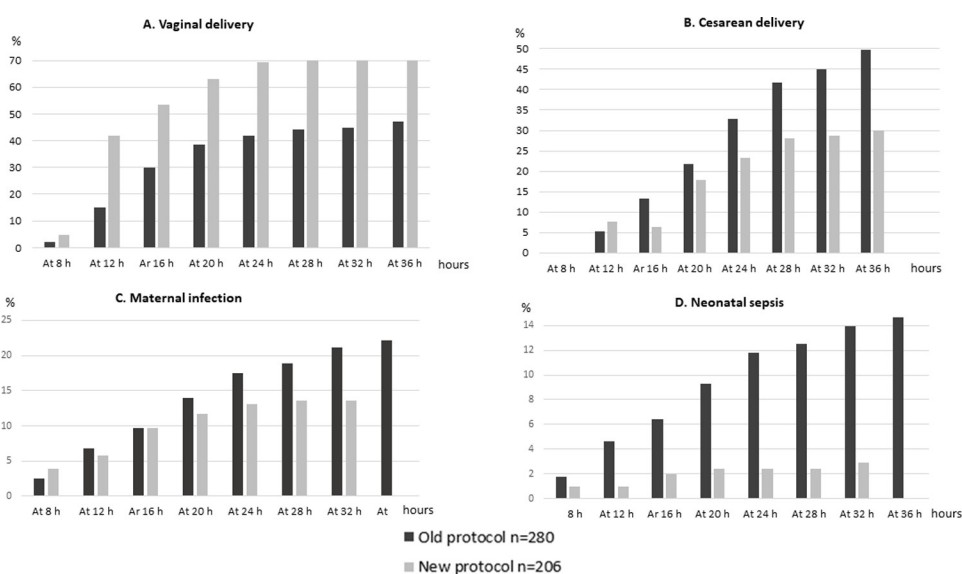

**Fig 3. The rates of vaginal delivery, cesarean delivery, and maternal and neonatal infections during the old and new oxytocin protocols.**

the new protocol, the rate of cesarean delivery in induced labor has decreased while the rate of labor induction is increasing [12]. A recent review by Dasanayakea and Goonewardene described a successful implementation of a new oxytocin protocol in a tertiary teaching hospital in 2006 with improved birth outcomes [13, 14], similar to our experience.

The standard regimen of oxytocin for women whose labor requires augmentation is started with 1–2 mU/minute dose, and increased over 30-minute intervals, the standard maximum dose is 32 mU/minute [3, 15]. The Cochrane review defines high-dose oxytocin regimen as infusion to deliver more than 100 mU oxytocin in the first 40 minutes and more than 600 mU

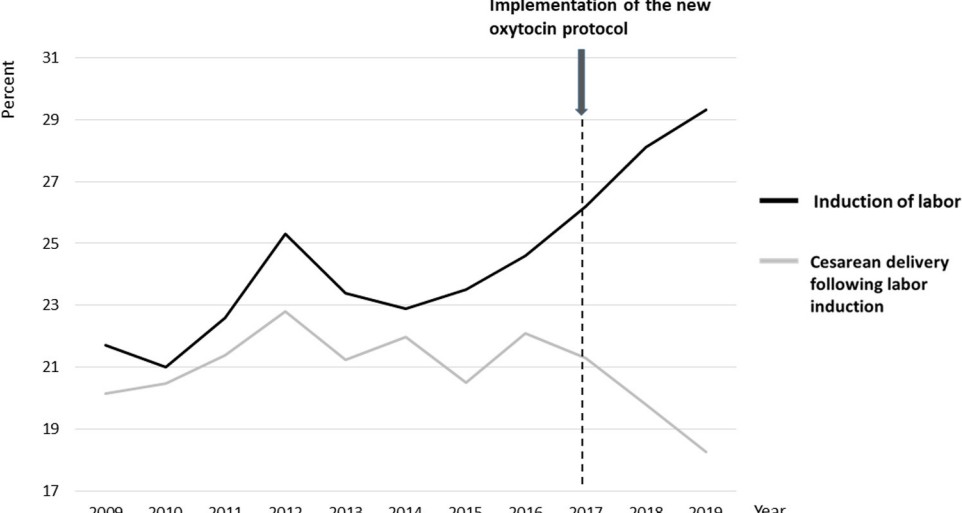

**Fig 4. The rates of labor induction and emergency cesarean delivery in induced labor in Helsinki University Hospital 2009–2019.** The rate of labor induction in Helsinki University hospital has steadily increased over the last decade. The rate of emergency cesarean delivery has decreased from 22.1% in 2016 to 18.3% in 2019 since the implementation of the new oxytocin protocol in January 2017 [12].

in two hours, and low dose regimen as infusion to deliver less than 100 mU oxytocin in the first 40 minutes and 600 mU in two hours [16]. In our study, in the old protocol 75 mU of oxytocin was infused during the first 40 minutes, and 600 mU during the first two hours. In the new protocol, 100 mU of oxytocin were infused during the first 40 minutes, and 800 mU during the first two hours. The maximum dose of the old protocol was 15 mU/ml with a total oxytocin dose of 7.6 IU over the 12-hour infusion, and the maximum dose of the new protocol was 20 mU/min with a total oxytocin dose of 12.38 IU over the 12-hour infusion.

The Cochrane review on spontaneous labor and oxytocin regimens concluded that high-dose oxytocin was associated with a reduction in the length of labor and caesarean section rate, and an increase in spontaneous vaginal delivery, but these associations did not remain significant after adjustment [4]. In the Cochrane review, or in the randomized trial on high and low dose oxytocin in induction of labor of 2391 women, no evidence was found that high-dose oxytocin increases either vaginal delivery within 24 hours or the caesarean section rate [5, 16]. Controversially, in our study the rates of cesarean section in both nulliparous and multiparous women were lower in the continuous high-dose regimen compared to the older low dose intermittent regimen, differently to the previous studies [16, 17]. Furthermore, we found a higher rate of women progressing into active phase of labor, and a shorter induction to delivery interval in the new high-dose protocol, as previously reported [18]. Discontinuation of oxytocin infusion in the active phase of labor, with cervical dilation of 6 cm and adequate uterine contractions, has been associated with lower risk of cesarean delivery and reduction of uterine tachysystole [19, 20]. This has been implemented in our hospital along with the new protocol. It is possible that all these oxytocin policy changes of the new protocol could have also contributed to the improved outcomes, in addition to the differences in oxytocin dosing.

It has been speculated, that prolonged exposure to oxytocin may desensitize oxytocin receptors, resulting in increased risk of postpartum hemorrhage [21, 22]. In our study, the rates of post-partum hemorrhage were similar in the short intermittent and continuous protocols, and no uterine ruptured occurred.

Cesarean delivery is recommended after approximately 15 hours of oxytocin infusion with no cervical change [23, 24]. After that an increase in adverse maternal and neonatal outcomes may occur, as seen in our study during the old protocol. In the new protocol, the decision on failed induction and cesarean delivery is performed at 12–18 hours from start of oxytocin [23–25]. However, it was noted that also during the new protocol, some women underwent prolonged induction of labor with duration of oxytocin induction of more than 18 hours against the hospital protocol. This may have been due to the newly implemented protocol that some obstetricians were not yet familiar or comfortable with.

The strength of this study is the setting in a tertiary academic hospital undergoing two different oxytocin protocols, and the detailed extensive patient data. The major weakness and bias of the study is the retrospective design. However, this study was a retrospective observational cohort on implementation of a new management protocol. The decision to include only women undergoing induction of labor at term, rather than at any time during the third trimester, is also a potential bias. We also regret not having the exact data on hyperstimulation. However, the rates of CTG changes and fetal distress leading to emergency cesarean delivery were recorded, and these rates were higher in the old protocol. Unfortunately, the data on CTG changes in labors leading to vaginal delivery were not recorded. Furthermore, we regret not having the data on maternal satisfaction during different protocols, as no trial has reported on maternal or caregiver satisfaction in the different oxytocin regimens. Another challenge with the study were also the relative terminology of "low dose" and "high dose" in studies on oxytocin for labor induction and augmentation, as well as the lack of standardized dosing. The authors acknowledge that the "high dose" of our new protocol still is lower than the dose used

in in the literature in this area, including the recent review on oxytocin protocols in 12 countries, the Cochrane review, and the new European oxytocin guideline [3, 16, 26]. Encouraged by these results and the recent literature, the authors are considering updating the maximum dose of our protocol higher, in line with the recent European guideline [26].

## Conclusion

Implementation of the new oxytocin protocol with a high starting dose of 140 mU/minute and continuous 12 (-18)-hour infusion with a maximum dose of 20 mU/minute, along with instituting a practice of stopping oxytocin once active labor is achieved and defining a maximum duration of induction before it was considered to have failed, resulted in higher rate of vaginal delivery and lower rate of maternal and neonatal infections compared to the previous low-dose oxytocin regimen. No difference in complications, such as post-partum hemorrhage, low 5-minute Apgar score, fetal asphyxia, or admission to neonatal intensive care unit were seen. Following implementation of the new oxytocin protocol, the rates of cesarean deliveries and maternal and neonatal infections in induced labor in our hospital have decreased. Our experience supports the use of high-dose continuous oxytocin induction regimen with a practice of stopping oxytocin once active labor is achieved, and a 15–18-hour maximum duration for oxytocin induction in the latent phase of labor following cervical ripening with a balloon catheter.

## Author Contributions

**Conceptualization:** Heidi Kruit.

**Data curation:** Leena Rahkonen.

**Formal analysis:** Leena Rahkonen.

**Investigation:** Heidi Kruit, Irmeli Nupponen.

**Methodology:** Leena Rahkonen.

**Project administration:** Heidi Kruit.

**Validation:** Irmeli Nupponen, Seppo Heinonen.

**Writing – original draft:** Heidi Kruit.

**Writing – review & editing:** Irmeli Nupponen, Seppo Heinonen, Leena Rahkonen.

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
