## [Decision Letter · Decision Letter 0]

26 Jan 2022

PONE-D-21-36119Comparison of delivery outcomes in low-dose and high-dose oxytocin regimens for induction of labor following cervical ripening with a balloon catheterPLOS ONE

Dear Dr. Kruit,

Thank you for submitting your manuscript to PLOS ONE. After careful consideration, we feel that it has merit but does not fully meet PLOS ONE’s publication criteria as it currently stands. Therefore, we invite you to submit a revised version of the manuscript that addresses the points raised during the review process.

We look forward to receiving your revised manuscript.

Kind regards,

Nnabuike Chibuoke Ngene, Dip HIV Med; MMed(FamMed); FCOG; MMed(O&G); Ph.D

Academic Editor

PLOS ONE

Journal Requirements:

2. Please provide additional details regarding participant consent. If you are reporting a retrospective study of medical records, archived samples or third party data, please ensure that you have discussed whether all data were fully anonymized before you accessed them.

Reviewers' comments:

Reviewer's Responses to Questions

**Comments to the Author**

1. Is the manuscript technically sound, and do the data support the conclusions?

Reviewer #1: Yes

Reviewer #2: Yes

2. Has the statistical analysis been performed appropriately and rigorously? 

Reviewer #1: I Don't Know

Reviewer #2: Yes

3. Have the authors made all data underlying the findings in their manuscript fully available?

Reviewer #1: No

Reviewer #2: No

4. Is the manuscript presented in an intelligible fashion and written in standard English?

Reviewer #1: Yes

Reviewer #2: Yes

5. Review Comments to the Author

Reviewer #1: General:

This study covers an interesting topic comparing the delivery outcomes of low-dose and high-dose oxytocin protocols for labour induction following cervical ripening by balloon catheter. Overall, the authors show that:

- The new high-dose protocol increases the rate of vaginal delivery.

- The new high-dose protocol reduces the rates of maternal and neonatal infection

- There was no difference between the rates of post-partum hemorrhage, umbilical artery blood pH-value <7.05 or neonatal intensive care admissions.

The study addresses how findings relate to previous research in the area. The introduction is supported by literature, it summarises the main controversies around oxytocin use (timing, dose infusion type, hyperstimulation, infection). Authors discuss limitations of the study.

Weaknesses: A weakness of the study is the lack of uterine hyperstimulation data, which authors also recognise as being a limitation of the study.

Suggested revisions

Major issues:

1. Title: It would be recommended for authors to add the study design in the title. For example: “Comparison of delivery outcomes in low-dose and high-dose oxytocin regimens for induction of labor following cervical ripening with a balloon catheter: A retrospective observational cohort study.”

2. Abstract:

a. Methods: Specify how many patients in each group i.e. low-dose oxytocin protocol (n= x) and high dose oxytocin protocol (n= y).

b. Results: Authors should add induction to delivery interval in results section of abstract as results from other secondary outcomes are stated.

3. Tables and figures:

a. Table 3: There are 206 patients in new protocol group in Table 3 as well as in Figure ,1 whereas the text states it is 207 patients. Adding a flow chart in the results section may be useful in clarifying this.

b. Table 4: Table 3 has total number of patients in each group at top of the table but table 4 does not.

c. Table 4: Authors should clarify the maternal infection value in old protocol (n=62) and subgroups intrapartum infection (n=46) and postpartum infection (n=18) as they do not add up. Furthermore, it might be useful to mention that the subgroups of maternal infection (intrapartum and postpartum infection) are not significantly different between the two protocols.

d. Table 4: It would be useful to also present CTG changes/fetal distress regardless of mode of delivery if data is available.

4. Methods:

a. If the authors have used the STROBE guidelines, please mention it in the methods. If it has not been used, please report this study in compliance with STROBE statement.

b. It would be recommended to state if any efforts were made to address potential risk of bias.

5. Results:

a. Could authors report numbers of individuals at each stage of study (examined for eligibility, confirmed eligible etc).

b. Page 14 lines 197-199: “In the old protocol, maternal or neonatal infection occurred in 29.3 % (n=82) of the labors, while in the new protocol the corresponding rate was 14.6 % (n=21); p<0.001”. This does not match results presented in table 4. In table 4 maternal or neonatal infection in the old protocol is reported in 82 patients (29.3%), and in 30 patients (14.6%) in the new protocol with a p value <0.0001.

c. Page 14 lines 201-203: The authors report that “The median birthweight, the rates of post-partum hemorrhage and umbilical artery blood pH-value <7.05 did not differ between the cohorts (Table 4).” However, in table 4 the mean SD birthweight is shown.

6. Discussion:

a. Page 17 lines 236-239 “The rate of vaginal delivery was higher, and the rates of maternal and neonatal infections were lower during the new protocol. After the implementation of the new protocol, the rate of cesarean delivery in induced labor has decreased while the rate of labor induction is increasing.” It would be useful for the authors to add p values.

7. Conclusion:

a. Page 20 lines 295-296: “No difference in complications, such as post-partum hemorrhage, or neonatal primary outcomes were seen.” It is not specified in the text what neonatal primary outcomes are, the primary outcome is defined as rate of vaginal delivery.

Minor issues:

8. Results page 3 line 24 and page 14 line 193: “p=0.004” whereas in table 4 it is p<0.004.

9. Page 11 lines 145-146 “Neonatal infections were by a neonatologist categorized...”. This could be rephrased to “Neonatal infections were categorized by a neonatologist...”.

10. Page 17 lines 239-240: There is a word missing “A recent review by described”

11. Figure 2: please add labels to axes

12. Figure 3: label years axis. I would also suggest adding a dotted line under the arrow to show more precisely when the new oxytocin protocol was implemented.

Reviewer #2: The manuscript compares delivery outcomes following an institutional change in practice from a low-dose oxytocin protocol to one with a higher dose. Overall the quality of the scientific work is very good and the outcomes discussed are important and add to the existing literature on this topic. The methods and results are explained very well and put into context quite clearly for the reader. I support acceptance of this manuscript with the following minor revisions:

1. I note that there were several changes made to the institutional protocol, in addition to the dose of oxytocin, which could potentially contribute to the differences between groups outlined in the results section. These included 1) removal of oxytocin breaks from the protocol 2) instituting a practice of stopping oxytocin once active labour is achieved and 3) instituting a maximum duration of induction before it was considered to have failed. It is possible that these policy changes could have also contributed to the improved outcomes the authors demonstrate in the new protocol, in addition to the differences in oxytocin dosing.

While these protocol changes are outlined by the authors in the Discussion section, it should be acknowledged that these may be confounding the study findings. The conclusion currently reads that the improved outcomes seen with their protocol change were the result of higher dose alone. I think that the Discussion could be modified slightly to highlight the possible contribution of these confounding factors and that the conclusion should reflect the change in protocol which includes but is not limited to a change from low dose to higher dose oxytocin. I don't think this takes away from the overall important findings of the paper.

2. The other point that might bear acknowledging in the manuscript is that "low dose" and "high dose" are relative terms in these types of studies. The "high dose" used here is lower than that used in other published studies. Lack of standardized dosing is one of the difficulties with the literature in this area, including the Cochrane review and other systematic reviews and meta analyses previously published. The findings here are specific to an institutional change from one protocol to another, which is important, particularly as it showed a clear benefit in terms the primary and secondary outcomes, but this finding may not be generalizable. I appreciate that the authors shared their protocol in its entirety, which would make adaptation by other institutions feasible.

The authors were not able to make all data publicly available due to restrictions from their Institutional Policy. They do provide contact information for those seeking access to the raw data, which I find acceptable.

Overall this is a strong paper and makes an important contribution to the existing literature. I did notice a few minor English grammatical errors, though the paper is overall well written and very clear. I congratulate the authors on their work.

6. PLOS authors have the option to publish the peer review history of their article (what does this mean?). If published, this will include your full peer review and any attached files.

Reviewer #1: No

Reviewer #2: **Yes: **Harrison Banner

---

## [Author Response · Author response to Decision Letter 0]

10 Mar 2022

Reviewer’s comments to the authors and authors’ response:

Reviewer #1 

Major issues:

1. Title: It would be recommended for authors to add the study design in the title. For example: “Comparison of delivery outcomes in low-dose and high-dose oxytocin regimens for induction of labor following cervical ripening with a balloon catheter: A retrospective observational cohort study.”

 Authors’s response: Thank you, this has been done.

2. Abstract:

a. Methods: Specify how many patients in each group i.e. low-dose oxytocin protocol (n= x) and high dose oxytocin protocol (n= y).

Authors’s response: Number of patients in each protocol have now been added in the methods section of the Abstract (lines 6-7).

b. Results: Authors should add induction to delivery interval in results section of abstract as results from other secondary outcomes are stated.

Authors’s response: The median (IQR) induction-to-delivery interval results have been added in the results section of the Abstract (lines 21-22). To still meet the limited 300-word count of the Abstract after these additions, we have slightly edited the abstract. All the removed words have been indicated with tracked changes.

3. Tables and figures:

a. Table 3: There are 206 patients in new protocol group in Table 3 as well as in Figure ,1 whereas the text states it is 207 patients. Adding a flow chart in the results section may be useful in clarifying this.

Authors’s response: There has been an unfortunate typo. The correct number of patients in the new protocol is 207, and this has now been corrected in Table 3 and Figure 2 (former Fig 1). We have also added a flow chart in the Results section (the new Fig 1).

b. Table 4: Table 3 has total number of patients in each group at top of the table but table 4 does not.

Authors’s response: The total number of patients has been added in each group at top of Table 4.

c. Table 4: Authors should clarify the maternal infection value in old protocol (n=62) and subgroups intrapartum infection (n=46) and postpartum infection (n=18) as they do not add up. Furthermore, it might be useful to mention that the subgroups of maternal infection (intrapartum and postpartum infection) are not significantly different between the two protocols.

Authors’s response: Maternal intrapartum infection occurred in 62 women and post-partum infection in 46 women in the old protocol. Two of the women who had an intrapartum infection presented with a separate post-partum cesarean wound infection later, a few days after being discharged from the hospital. Both women had antibiotic treatment and revision surgery. This has now been clarified in the footnotes of Table 4. 

d. Table 4: It would be useful to also present CTG changes/fetal distress regardless of mode of delivery if data is available.

Authors’s response: The authors regret that we don’t have the data on hyperstimulation as discussed in the limitations of the study. CTG changes and fetal distress leading to emergency cesarean delivery have been recorded, however, unfortunately the data on CTG changes in women who had vaginal delivery were not recorded. We have now emphasized this in the limitations paragraph of the Discussion (lines 282-286).

4. Methods:

a. If the authors have used the STROBE guidelines, please mention it in the methods. If it has not been used, please report this study in compliance with STROBE statement.

Authors’s response: The STROBE guideline has been used, and this is now stated in the Methods (lines 81-82). 

b. It would be recommended to state if any efforts were made to address potential risk of bias.

Authors’s response: The authors acknowledge the potential bias being the retrospective design and the decision to include only women undergoing induction of labor at term, rather than at any time during the third trimester. We tried to address this bias by choosing two identical cohorts within two identical time periods during the two different oxytocin protocols. This has been discussed in the limitation of the study (lines 278-282), we now added it in the Methods as well (lines 71-74). 

5. Results:

a. Could authors report numbers of individuals at each stage of study (examined for eligibility, confirmed eligible etc).

Authors’s response: We identified 3290 and 3085 deliveries during the selected time periods of the old and new protocol, examined 743 and 853 women for eligibility, and confirmed 280 and 207 women eligible, respectively. This has now been added in the beginning of the Results (157-159). We have also added the flowchart (Fig 1). 

b. Page 14 lines 197-199: “In the old protocol, maternal or neonatal infection occurred in 29.3 % (n=82) of the labors, while in the new protocol the corresponding rate was 14.6 % (n=21); p<0.001”. This does not match results presented in table 4. In table 4 maternal or neonatal infection in the old protocol is reported in 82 patients (29.3%), and in 30 patients (14.6%) in the new protocol with a p value <0.0001.

Authors’s response: Thank you for pointing this out, the mistake (n=21 instead of n=30) and p> 0.001 in the the Results and Table 4 have been corrected (line 190). 

c. Page 14 lines 201-203: The authors report that “The median birthweight, the rates of post-partum hemorrhage and umbilical artery blood pH-value <7.05 did not differ between the cohorts (Table 4).” However, in table 4 the mean SD birthweight is shown.

Authors’s response: Thank you, this sentence has now been corrected, the birthweight is expressed as mean (SD), not median (line 192). 

6. Discussion:

a. Page 17 lines 236-239 “The rate of vaginal delivery was higher, and the rates of maternal and neonatal infections were lower during the new protocol. After the implementation of the new protocol, the rate of cesarean delivery in induced labor has decreased while the rate of labor induction is increasing.” It would be useful for the authors to add p values.

Authors’s response: We have now added the p-values and the rates to this sentence in the Discussion (lines 229-231).

7. Conclusion:

a. Page 20 lines 295-296: “No difference in complications, such as post-partum hemorrhage, or neonatal primary outcomes were seen.” It is not specified in the text what neonatal primary outcomes are, the primary outcome is defined as rate of vaginal delivery.

Authors’s response: This has been rephrased: “No difference in complications, such as post-partum hemorrhage, low 5-minute Apgar score, fetal asphyxia or admission to neonatal intensive care unit were seen” (lines 302-304).

Minor issues:

8. Results page 3 line 24 and page 14 line 193: “p=0.004” whereas in table 4 it is p<0.004.

Authors’s response: Thank you, this has been corrected

9. Page 11 lines 145-146 “Neonatal infections were by a neonatologist categorized...”. This could be rephrased to “Neonatal infections were categorized by a neonatologist...”.

Authors’s response: We have rephrased this as suggested (lines 134-135).

10. Page 17 lines 239-240: There is a word missing “A recent review by described”

Authors’s response: Thank you, this has been corrected as “A recent review by Dasanayakea and Goonewardene described..” (lines 233-234).

11. Figure 2: please add labels to axes

Authors’s response: The axes of Fig 2 have now been labeled

12. Figure 3: label years axis. I would also suggest adding a dotted line under the arrow to show more precisely when the new oxytocin protocol was implemented.

Authors’s response: We have added label to the x-axis and a dotted line under the arrow in Fig 3.

Reviewer #2: 

Minor revisions:

1. I note that there were several changes made to the institutional protocol, in addition to the dose of oxytocin, which could potentially contribute to the differences between groups outlined in the results section. These included 1) removal of oxytocin breaks from the protocol 2) instituting a practice of stopping oxytocin once active labour is achieved and 3) instituting a maximum duration of induction before it was considered to have failed. It is possible that these policy changes could have also contributed to the improved outcomes the authors demonstrate in the new protocol, in addition to the differences in oxytocin dosing.

While these protocol changes are outlined by the authors in the Discussion section, it should be acknowledged that these may be confounding the study findings. The conclusion currently reads that the improved outcomes seen with their protocol change were the result of higher dose alone. I think that the Discussion could be modified slightly to highlight the possible contribution of these confounding factors and that the conclusion should reflect the change in protocol which includes but is not limited to a change from low dose to higher dose oxytocin. I don't think this takes away from the overall important findings of the paper.

Authors’s response: Thank you for this excellent comment. We have now highlighted these confounding factors as a part of the new protocol in the Discussion and in the Conclusion (Discussion lines 226-229 and 261-264, Conclusion lines 298-300 and 307-309).We also edited the conclusion paragraph of the Abstract accordingly (lines 24-28).

2. The other point that might bear acknowledging in the manuscript is that "low dose" and "high dose" are relative terms in these types of studies. The "high dose" used here is lower than that used in other published studies. Lack of standardized dosing is one of the difficulties with the literature in this area, including the Cochrane review and other systematic reviews and meta analyses previously published. The findings here are specific to an institutional change from one protocol to another, which is important, particularly as it showed a clear benefit in terms the primary and secondary outcomes, but this finding may not be generalizable. I appreciate that the authors shared their protocol in its entirety, which would make adaptation by other institutions feasible.

Authors’s response: Thank you for pointing this out. The authors acknowledge our “high” oxytocin dose still remains lower compared to several other institutions, the Cochrane review and the recent European guideline. However, at the time of implementation of the new protocol in Finland, our current dose was considered extremely high compared to other national institutions. We have now discussed this topic and the lack of standardized dosing in the Discussion (lines 288-295). Encouraged by the results of this study, we are considering updating the maximum oxytocin dose of our institutional protocol higher, similar to the recommendation in the current European guideline (lines 293-295).

We thank the editor and reviewers for these valuable comments which have much improved the manuscript.

---

## [Decision Letter · Decision Letter 1]

8 Apr 2022

Comparison of delivery outcomes in low-dose and high-dose oxytocin regimens for induction of labor following cervical ripening with a balloon catheter: a retrospective observational cohort study

PONE-D-21-36119R1

Dear Dr. Kruit,

We’re pleased to inform you that your manuscript has been judged scientifically suitable for publication and will be formally accepted for publication once it meets all outstanding technical requirements.

Kind regards,

Nnabuike Chibuoke Ngene, Dip HIV Med; MMed(FamMed); FCOG; MMed(O&G); Ph.D

Academic Editor

PLOS ONE

Additional Editor Comments (optional):

Reviewers' comments:

Reviewer's Responses to Questions

**Comments to the Author**

1. If the authors have adequately addressed your comments raised in a previous round of review and you feel that this manuscript is now acceptable for publication, you may indicate that here to bypass the “Comments to the Author” section, enter your conflict of interest statement in the “Confidential to Editor” section, and submit your "Accept" recommendation.

Reviewer #2: All comments have been addressed

2. Is the manuscript technically sound, and do the data support the conclusions?

Reviewer #2: Yes

3. Has the statistical analysis been performed appropriately and rigorously? 

Reviewer #2: Yes

4. Have the authors made all data underlying the findings in their manuscript fully available?

Reviewer #2: No

5. Is the manuscript presented in an intelligible fashion and written in standard English?

Reviewer #2: No

6. Review Comments to the Author

Reviewer #2: (No Response)

7. PLOS authors have the option to publish the peer review history of their article (what does this mean?). If published, this will include your full peer review and any attached files.

Reviewer #2: No

---

## [Editor Report · Acceptance letter]

13 Apr 2022

PONE-D-21-36119R1 

Comparison of delivery outcomes in low-dose and high-dose oxytocin regimens for induction of labor following cervical ripening with a balloon catheter: a retrospective observational cohort study 

Dear Dr. Kruit:

I'm pleased to inform you that your manuscript has been deemed suitable for publication in PLOS ONE. Congratulations! Your manuscript is now with our production department. 

Kind regards, 

on behalf of

Dr. Nnabuike Chibuoke Ngene 

Academic Editor

PLOS ONE